# Tinnitus and Neuropsychological Dysfunction in the Elderly: A Systematic Review on Possible Links

**DOI:** 10.3390/jcm10091881

**Published:** 2021-04-27

**Authors:** Rita Malesci, Francesca Brigato, Tiziana Di Cesare, Valeria Del Vecchio, Carla Laria, Eugenio De Corso, Anna Rita Fetoni

**Affiliations:** 1Audiology Section, Neuroscience, Reproductive Sciences and Dentistry Department, “Federico II” University, via Pansini 5, 80131 Naples, Italy; rita.malesci@unina.it (R.M.); valeria.delvecchio@unina.it (V.D.V.); carla.laria@unina.it (C.L.); 2Department of Otolaryngology Head & Neck Surgery, School of Medicine, Università Cattolica del Sacro Cuore, Largo F. Vito 1, 00168 Rome, Italy; franscescabrigato13@gmail.com (F.B.); Tizianadicesare90@gmail.com (T.D.C.); eugenio.decorso@policlinicogemelli.it (E.D.C.); 3Fondazione Policlinico Universitario A. Gemelli IRCCS, Largo A. Gemelli 8, 00168 Rome, Italy

**Keywords:** tinnitus, cognition, psychological aspects, auditory pathways, elderly

## Abstract

Introduction: Tinnitus is a common and disabling symptom often associated with hearing loss. While clinical practice frequently shows that a certain degree of psychological discomfort often characterizes tinnitus suffers, it has been recently suggested in adults as a determining factor for cognitive decline affecting attention and memory domains. The aim of our systematic review was to provide evidence for a link between tinnitus, psychological distress, and cognitive dysfunction in older patients and to focus on putative mechanisms of this relationship. Methods: We performed a systematic review, finally including 192 articles that were screened. This resulted in 12 manuscripts of which the full texts were included in a qualitative analysis. Results: The association between tinnitus and psychological distress, mainly depression, has been demonstrated in older patients, although only few studies addressed the aged population. Limited studies on cognitive dysfunction in aged patients affected by chronic tinnitus are hardly comparable, as they use different methods to validate cognitive impairment. Actual evidence does not allow us with certainty to establish if tinnitus matters as an independent risk factor for cognitive impairment or evolution to dementia. Conclusion: Tinnitus, which is usually associated with age-related hearing loss, might negatively affect emotional wellbeing and cognitive capacities in older people, but further studies are required to improve the evidence.

## 1. Introduction

Tinnitus consisting of the perception of sounds in the absence of external stimuli is a very common and disabling condition with pervasive effects on health and wellbeing. In 95% of cases, tinnitus is subjective and described as a buzz, click, hiss, ring, roar, hum, or pulsatile. The prevalence increases with age, affecting 24–45% of the elderly [1], where tinnitus is frequently associated to hearing impairment. Noise exposure, which is the greater risk factor for the development of both hearing loss and tinnitus [2,3], contributes to the increase in prevalence among the elderly. Tinnitus can occur as an isolated idiopathic symptom or in association with otologic disease, such as otosclerosis [4] and Meniere disease [5], drug ototoxicity [6], cerebrovascular diseases, hypertension, dyslipidemia, metabolic diseases [7], chronic kidney disease, and diabetes mellitus [8]. On the other hand, it could be associated with any type of deafness as well as with a normal hearing threshold as in the case of ototoxicity from aspirin and quinine [9] or migraine [10]. Cochlear mechanism and involvement of central auditory and non-auditory pathways thought to underlie tinnitus with or without hearing loss are still controversial. Major evidence suggests that tinnitus is related to a failure of the central auditory pathway to adapt to the loss of afferent peripheral fibers due to peripheral damage [11,12], leading to plastic neuronal changes in the tonotopic map of the auditory cortex, as a “maladaptive plasticity”, which concurs in maintaining tinnitus in a sort of “vicious circle” [13,14,15,16,17]. All these changes in the central auditory pathway, together with the neuroplastic reorganization within the thalamus and the structures of the limbic and paralimbic circuits [18], induced us to speculate about a possible relationship between tinnitus, psychological distress, and cognitive impairment, with a positive correlation to tinnitus severity [19,20]. Although the higher prevalence of tinnitus in older adults is associated with hearing impairment, insomnia, depression, anxiety, and cognitive dysfunction, there are only a few studies addressing a link and causal mechanism, especially in the elderly. Thus, our aim is to provide evidence for a relationship between tinnitus and psychological distress or cognitive dysfunction in older patients through a systematic review of the literature.

## 2. Methods

The systematic review was conducted, accordingly with our previous study [21], following the Preferred Reporting Items for Systematic Review and Meta-Analysis process (PRISMA) [22], to identify clinical studies about tinnitus and cognitive decline or tinnitus and psychological disorders in the elderly. Manuscripts published from January 2000 to January 2021 were screened primarily by Ovid Medline (Wolters Kluwer, New York City, NY, USA) and EMBASE (Elsevier, Amsterdam, The Netherland) and from other sources (PubMed Central (National Center for Biotechnology Information, Bethesda, MD, USA), Cochrane review (Cochrane, London, England), Web of Science (Clarivate Analytics PLC, Philadelphia, PA, USA), and Google Scholar (Google, Mountain View, CA, USA)). Literature searches were performed in January 2021.

### 2.1. Study Selection

We went through two different searches using MeSH terms. One group of Authors focused on the relationship between chronic tinnitus and cognitive decline, matching the term as follow: [(Tinnitus)] AND [(cognitive decline) OR (cognitive impairment)] AND [(ageing) OR (elderly) OR (older people)]. The second group focused on studies about chronic tinnitus and psychological disorders, matching the term as follow: [(Tinnitus)] AND [(depression) OR (anxiety) OR (psychological disorders)] AND [(ageing) OR (elderly) OR (older people)]. Manuscripts were screened by PubMed. Firstly, authors read the articles’ titles and abstracts and selected those interesting as being inclusive as possible. The inclusion criteria were primary research studies (including those descriptive and observational, randomized trials, and basic science articles) published after January 2000 on tinnitus in the elderly and its association with cognitive decline and psychological alterations. Only population studies >50 years were considered. We excluded those that did not join the inclusion criteria or deal directly with the issue investigated; particularly, we excluded all the articles that referred to childhood or young adulthood. Only full text available articles were included. We considered only English-language peer-reviewed papers.

### 2.2. Qualitative Analysis

All included studies were evaluated for quality (level of evidence IV-III-II-I) based on study design, randomization, measure outcome reliability, and number of patients enrolled. Each author individually scored studies, and quality was assigned only after unanimous consensus.

## 3. Results

Overall, our search generated 192 articles after duplicates exclusion. We removed 45 articles due to publication time and article type, as above reported. This resulted in 147 publications of which full texts were assessed. We excluded 135 articles that do not meet the inclusion criteria or deal directly with the issue investigated. In total, our systematic search provided 12 articles. Details of the search performed are shown in the flowchart in Figure 1. Tables summarize the included studies (Table 1 and Table 2).

### 3.1. Psychological Distress

It is largely demonstrated that psychiatric discomfort is present in a large number of tinnitus suffers, with a higher prevalence of anxiety rather that depression [35,36] (Table 1). Tinnitus may directly determine a psychiatric condition; even if sleep disorders and insomnia evoked by tinnitus could induce emotional distress or unmask a pre-existing but compensated disorder. Aazh H et al. [24] retrospectively verified a strong association between tinnitus annoyance, depression level, and insomnia score in the elderly. On the other hand, the appearance of a psychiatric disease may worsen previously well-tolerated tinnitus. It has been recently demonstrated in our series a high prevalence of psychological comorbidities (i.e., in about 60% of patients) among tinnitus adult patients [35]. Furthermore, our recent study in older people [23] demonstrated that the subjective perception of tinnitus discomfort as measured with the Tinnitus Handicap Inventory (THI) score was strongly related to psychological distress, whereas there was no relationship between tinnitus severity and cognitive dysfunction detected using the Mini Mental State Examination (MMSE). Interestingly, recent literature has showed that tinnitus patients do not usually develop major depressive disorders, but mild psychiatric symptoms, leading slowly to impaired quality of life [37]. On the contrary, large longitudinal cohort studies [26,27] demonstrated a worsening of quality of life and psychological wellbeing in those elderly subjects experiencing tinnitus. Park et al. [30] showed significantly higher rates of depressive mood, psychological distress, and suicidal ideations in tinnitus suffers ≥65 years old compared to healthy controls. Some authors [28] sustain a significant positive association between depression and tinnitus being at least a moderate problem; according to them, it would be worth investigating the possible coexistence of anxiety and depression only in moderate or severe tinnitus. Consequently, the use of tools allowing staging of the severity of tinnitus is essential. Furthermore, the routine administration of simple and reliable screening tests for dysfunctional psychological traits addressed to non-psychiatric patients could be useful in patients with moderate to severe tinnitus having higher risk of developing anxious–depressive symptoms. It could be that elderly subjects are more exposed to the development of anxiety and depression than younger tinnitus suffers due to the physical fragility and social isolation that frequently involve older people. Conversely, a study, which aimed to explore the differences in various tinnitus-related features between the younger and older adult patients, did not find relevant difference in depressive symptoms and the stress levels between groups [25]. Considering the high association between tinnitus annoyance and emotional wellbeing, some authors [29] recently proposed the use of an internet-based cognitive behavioral therapy intervention to reduce tinnitus severity finding a significant reduction in tinnitus distress and comorbidities (i.e., insomnia, depression, cognitive failures) and a significant improvement in quality of life, confirming the close relationship between tinnitus severity and emotional disturbance. Taken together, results in the literature confirm the association between tinnitus severity and psychological distress and reduced quality of life in older patients.

### 3.2. Cognitive Impairment

Based on behavioral evidence, psychological markers of attention switching (i.e., cognitive and emotional control) are impaired in chronic tinnitus suggesting that the reduced cognitive control may be pivotal in maintaining the awareness of tinnitus. According to the literature (Table 2), tinnitus patients have poor cognitive performance, but it is still unclear whether cognitive impairment is a response to tinnitus manifestations or a feature of it, especially in the elderly. By using MMSE as a screening tool for cognitive impairment hearing threshold and anxious–depressive traits measured with hospital anxiety depression scale (HADS) questionnaire, scores for cognitive dysfunction were slightly increased by age and gender [23] (Table 1). Few other reports suggest the relationship between neurocognitive abilities and tinnitus severity [38,39], even if its mechanism remains controversial [5]. Thus, a key point in the elderly population is whether cognitive impairment is related to age-related hearing loss (ARHL) or to tinnitus per se. Indeed, Lee et al. [32] showed that, adjusting for age, sex, and hearing threshold, patients older than 65 years with a THI score greater than 30 were affected by mild cognitive impairment (MCI). However, conclusions of the study had some limitations; in fact, the hearing threshold was worse in the MCI, group showing a stronger relationship with cognitive deficit than the THI score. For now, there are no studies on older tinnitus suffers with normal hearing that could clarify this critical issue. Again, hearing loss has been reported to be an independent risk factor for dementia; the link between ARHL and cognitive impairment is still understood [23,36]. Interestingly, it has been demonstrated that the reduction in c-proteasome plasma activity as a specific marker of cognitive decline in patients with chronic tinnitus [33] predicted its accumulation in cells with a similar pattern of amiloid-β protein in Alzheimer’s disease. Older tinnitus sufferers that underwent behavioral cognitive therapy showed an improvement in cognitive abilities as well as in tinnitus severity [29]. A complex age-related clinical condition has been recently studied by Ruan et al. [34], which reported an association between frailty, cognitive impairment, and chronic tinnitus. Frailty is a heterogeneous clinical condition characterized by a vulnerability to stressors. The coexistence of cognitive impairment and frailty are known as cognitive frailty. These Authors measured MCI using dedicated tests for every cognitive domain (executive, attention, memory, language, etc.), adjusting results for comorbidities (i.e., cardiovascular disease, diabetes), age, sex, smoking, and alcoholic use, and demonstrated that severe tinnitus was associated with cognitive frailty but not with physical frailty. How chronic tinnitus causes cognitive deficits might be found in functional and structural brain alterations that were studied by Lee et al. [31]. Patients with MCI were divided in two groups (tinnitus and non-tinnitus groups) and were tested through FDG-PET to evaluate glucose metabolic connectivity. The tinnitus–MCI group showed a lower metabolism in the right superior temporal pole (which comprises the auditory cortex and is associated with the cognitive social processes interacting with limbic areas) and in the fusiform gyrus (which is altered in semantic dementia), if compared with non-tinnitus group. Furthermore, they exhibited significantly lower gray matter volume in the right insula (which is involved in the emotional reaction to tinnitus), and the THI was inversely correlated with it. Thus, the evidence for a causal link between tinnitus and MCI might be found in the pathophysiology of tinnitus and in the central neural changes that it determines. Surely, future studies in this direction could help to establish the causal link.

## 4. Discussion

As given above, the association between tinnitus and psychological distress has been demonstrated, although only a few studies addressed this topic to the elderly. More difficult is the task of proving the association between cognitive dysfunction and tinnitus. Patients with chronic tinnitus (with or without hearing impairment) refer attention and memory failures, which reflects the dropped ability to shift attention away from phantom sounds in order to achieve proper cognitive performances [39]. However, it is still lacking evidence for precise cognitive constructs, which might determine effects on mnemonic and attentive domains in older patients. Many questionnaires to assess the tinnitus severity and its effect on emotional and cognitive domains (such as cognition, emotion, sleep, communication, and quality of life) have been used; therefore, it is difficult to compare results. Furthermore, objective measures and tests validated for patients with hearing impairments are still lacking. Existing self-administered screening tests for psychological distress and cognitive performance in association with the well-known THI are helpful tools for the initial assessment of tinnitus and the monitoring during treatment [24,40]. Therefore, the use of screening questionnaires will be helpful to address older patients to adequate management, assessing who could be at risk for cognitive dysfunction and development of dementia.

A current key point is to evaluate if tinnitus-related comorbidities effectively depend on tinnitus or the hearing loss that is often associated. Limitations of this systematic review are that there are no reports on the effect of tinnitus on specific cognitive domains and its impact in older patients with good hearing. Furthermore, a major concern is that the definition of tinnitus could be ambiguous in addressing phantom sound perception related to hearing dysfunction alone, and the other symptoms may be underestimated. Considering the higher presence of comorbidities, as recently suggested, the associated symptoms including emotional distress and cognitive dysfunction can be more correctly expressed by the definition of “tinnitus disorder” [41]. We believe that this may explain the poor evidence found between cognitive dysfunction and tinnitus as a limitation of this systematic review; thus, further studies are desirable in focusing disorders related to tinnitus. After all, it is known the relationship between functional effects and hearing deprivation, especially in older patients, affecting the morphology and function of specific brain regions [42]. In fact, as is well known based on the theory of the cognitive load, effortful hearing induces a contraction of available cognitive resources recruiting regions in the frontal and temporo-parietal cortex [43] leading to reduced cognitive reserve, predisposing to cognitive decline [44]. On the other hand, this link is still understudied for tinnitus disorders.

Another limit is that, even if some studies (i.e., those of Lee and Yun) evaluated the relationship with MCI, which is a common dysfunction in older patients [45], results are hardly comparable, as they use different methods to validate cognitive impairment. Therefore, it is hard to understand if tinnitus matters as an independent risk factor. However, we may assume that tinnitus, which is usually associated with ARHL, might negatively affect cognitive function or contribute to the evolution of MCI to dementia. The pathogenic mechanisms involved in the ARHL are considered common in tinnitus onset and comprehend sensorial deprivation and cognitive reserve reduction, supported by a shared pathological pathway (such as a microvascular damage of the brain). Hearing loss diverts attention to auditory processes, weakening executive controls, which are physiologically declined by age, therefore holding a vicious circle in which memory, praxis, and language are affected [46,47]. Individuals with both tinnitus and hearing loss have more severe reactions to tinnitus than those with normal hearing [48,49]. However, age-related decline is associated with the impairment of mental functions, as well as starting from middle age or even earlier, attention, memory, executive functions, and processing speed are affected [50,51,52,53,54]. Therefore, concomitant hearing loss and tinnitus interfere with the age-related dysfunction through the involvement of the same auditory and non-auditory networks [55]. A distinct contribution of tinnitus in cognitive impairment as well as in the risk of dementia is still lacking. Further functional and neuroimaging studies will be desirable in the explanation of the emotional and cognitive aspects of tinnitus. With greater knowledge of the neuronal mechanisms of tinnitus and related comorbidities, new approaches for accurate diagnosis and effective therapy could be explored.

## 5. Conclusions

Tinnitus is a common and disabling symptom especially in adults, and it is often associated with hearing loss in older people. A systematic review of the literature confirms a link between tinnitus severity and psychological distress. Elderly patients affected by chronic tinnitus have dysfunctional traits, such as anxiety and depression, and demonstrate reduced cognitive functions. In older patients affected by hearing loss, tinnitus seems to worsen cognitive dysfunction. However, further studies are required to improve the evidence supporting the relationship between cognitive dysfunction and tinnitus.

## Figures and Tables

**Figure 1 jcm-10-01881-f001:**
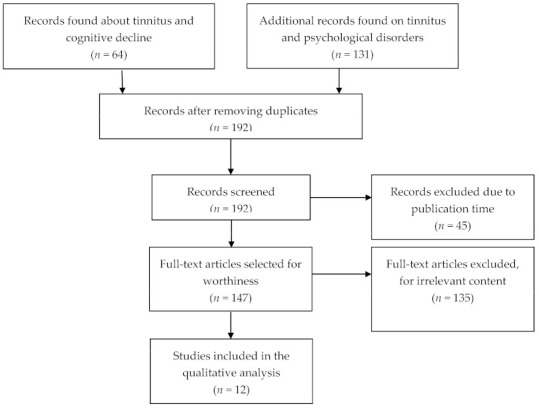
PRISMA flow diagram. The flowchart displays article search and selection.

**Table 1 jcm-10-01881-t001:** Evidence linking chronic tinnitus to psychological disorders in the elderly.

Author, Year [Ref]	Inclusion Criteria	Exclusion Criteria	N° of Cases, Age	Type of Study	Objective	Methods (Outcome Evaluation)	Results	Level of Evidence	Evidence of Association
Fetoni AR, 2021 [23]	-Chronic tinnitus -Age ≥ 55 y -With or without HL	-History of neurological diseases -Psychiatric disorders -Otologic diseases-Antipsychotic drugs use	102 patients ≥55 y; 70.4 ± 9.6 y (range 55–94 y	Prospective cross-sectional study	To assess the value of self-administered screening tests in comparing severity of tinnitus perception with emotional disorders and cognitive status	THI, HADS, MMSE (questionnaires)	THI score related to HADS-A score, HADS-D score, there was no relationship between tinnitus severity and MMSE	II	Yes
Aazh H, 2017 [24]	-Age ≥ 60 y -Tinnitus sufferers with/without hyperacusis -With/without HL	History of neurologic and psychiatric diseases, or sleep disorders	184 patients ≥60 y; mean age of 69 y	Retrospective cross-sectional study	To assess issues associated with tinnitus and hyperacusis handicap	HADS, HQ, ISI, THI, VAS (questionnaires)	THI was significant in predicting tinnitus annoyance. Hyperacusis handicap and insomnia were both predicted by level of depression	III	Yes
Park SY, 2017 [25]	-Age < and ≥ 65 y -With/without HL	History of psychiatric or neurologic diseases	76 patients ≥65 y	Retrospective cohort study	To assess differences between tinnitus-related features and psychological aspects between younger and older tinnitus sufferers	THI, BDI, and BEPSI (questionnaires)	No differences in tinnitus severity, depression, and stress between younger and older subjects	III	Yes/No
Gopinath B, 2010 [26]	-Age ≥ 50 y -With/without tinnitus -With/without HL	History of psychiatric diseases	1214 participants (602 with tinnitus) ≥50 y	Longitudinal cohort study	To evaluate the risk factors and effects of tinnitus (depression)	SF-36; MHI for depression and/or CES-D (questionnaires)	Lessened quality of life and wellbeing in patients with tinnitus	II	Yes
Lasisi AO, 2010 [27]	-Age ≥ 65 y -With/without tinnitus; -With/without HL	History of neuropsychologic diseases	184 patients ≥65 y; mean age of 77.3 y	Longitudinal cohort study	To establish the prevalence of tinnitus in the elderly and its influence on their life quality	WHOQOL (questionnaire)	Tinnitus sufferers had a lower perception of their overall health and a worse life quality; twofold likelihood to suffer activities impairment in daily living	II	Yes
Loprinzi PD, 2013 [28]	-Age ≥ 70 y -With/without tinnitus -With/without HL	Age < 70 years old	696 patients 76 ± 0.2 y (range 70–85 y)	Prospective cross-sectional study	To evaluate the association between tinnitus and depression	Patient Health Questionnaire-9 (questionnaire)	Positive association between tinnitus (at least a moderate) and depression; patients bothered by tinnitus before going to bed were nearly 3 times more likely to be depressed	II	Yes, in moderate/severe tinnitus
Beukes EW, 2018 [29]	-Age< and > 60 y -Chronic tinnitus -With/without HL who completed therapy	History of neuropsychiatric diseases	146 patientsstratified for age (> 60 y)	A randomized, delayed intervention efficacy trial	To establish if an internet-based cognitive behavioral therapy is useful in reducing tinnitus severity and associated comorbidities	ISI, Generalized Anxiety Disorder-7, Patient Health Questionnaire-9, Hearing Handicap Inventory for adults, HQ, Cognitive Failures Questionnaire, Satisfaction with Life Scales (questionnaires	Significant reduction in tinnitus and comorbidities (insomnia, depression, hyperacusis, cognitive failures) and improving of life quality	I	Yes
Park HM, 2020 [30]	-Age ≥ 60 y -With/without tinnitus -With/without HL	History of neurologic or psychiatric diseases	5129 patients ≥60 y (range 60–79 y)	Retrospective cross-sectional study	To find a possible association of tinnitus, mental health, and health-related quality of life	Stress was tested asking: “How much stress do you usually feel in your daily life?”; depression was tested using CIDI-SF; suicide ideation with the question: “Have you ever thought about committing suicide within 12 months?” (questionnaires)	The annoying tinnitus patients had more depression, psychological stress, and suicidal ideation if compared to control group	III	Yes

y = years; HL = hearing loss; THI = Tinnitus Handicap Inventory; HADS = Hospital Anxiety and Depression Scale; MMSE = Mini Mental State Examination; HQ = Hyperacusis Questionnaires; ISI = Insomnia Severity Index; BDI = Beck’s Depression Inventory; BEPSI = Brief Encounter Psychosocial Instrument; SF-36 = Short Form 36-item Health Survey; MHI = Mental Health Index; CES-D = Centre for Epidemiologic Studies Depression Scale; WHOQOL = World Health Organization Quality of Life; CIDI-SF = World Health Organization’s Composite International Diagnostic Interview-Short Form.

**Table 2 jcm-10-01881-t002:** Evidence linking chronic tinnitus to cognitive decline in the elderly.

Author, Year [Ref]	Inclusion Criteria	Exclusion Criteria	N° of Cases, Age	Type of Study	Objective	Methods (Outcome Evaluation)	Results	Level of Evidence	Evidence of Association
Lee SY, 2020 [31]	-Age > 60 y -Diagnosed for MCI -With/without tinnitus	-Moderate or severe hearing loss -Otologic diseases -History of psychiatric or neurologic disorders	23 patients (12 with tinnitus) 74.0 ± 6.1 y (range 63–83 y)	Retrospective cohort study	To examine the glucose metabolism and gray matter volume in patients with MCI and tinnitus	MCI with or without tinnitus: FDG-PET and magnetic resonance imaging were performed (imaging)	Specific brain regions are associated with cognitive decline and increased tinnitus severity	III	Yes
Lee SY, 2020 [32]	-Age ≥ 65 y -With/without HL	History of psychiatric or neurologic disorders	58 patients68.1 ± 5.1 y (range 65–82 y)	Prospective cohort study	To examine the cognitive domains and the association between tinnitus severity and cognitive functions	K-PHQ-9, K-IADL, MoCA-K (questionnaires)	THI score in the MCI group was higher than in the non-MCI	II	Yes
Yun Y, 2020 [33]	-Age > 50 y -Chronic tinnitus	-History of Alzheimer’s disease or neurologic diseases -Hearing loss or otologic diseases	55 patients > 50 y	Cross-sectional study	To examine plasma c-proteasome activity in association with cognitive functions in chronic tinnitus patients	Plasma c-proteasome activity was achieved with fluorogenic reporter substrate; MoCA (cut-off score of 22/23) to assess MCI (markers and questionnaires)	Circulating proteasomes were lower in patients with chronic tinnitus and MCI	III	Yes
Fetoni AR, 2021 [23]	-Chronic tinnitus -Age ≥ 55 y -With/without HL	-History of neurological diseases -Psychiatric disorders -Otologic diseases-Antipsychotic drugs use	102 patients ≥ 55 y	Prospective cross-sectional study	To evaluate the use of self-administered screening tests to correlate the severity of tinnitus with emotional disorders and the overall cognitive status	THI, HADS, MMSE (questionnaires)	THI score was directly related to HADS score, there was no relationship between tinnitus severity and MMSE	II	No
Beukes EW, 2018 [29]	-Age < and > 60 y -Chronic tinnitus-With/without HL -Who completed therapy	History of neuropsychiatric diseases	146 patients stratified for age (>60 y)	Randomized delayed intervention efficacy trial	To establish if an internet-based cognitive behavioral therapy is useful to lessen tinnitus severity and associated comorbidities	ISI, Generalized Anxiety Disorder, Patient Health Questionnaire, Hearing Handicap Inventory for Adults Screening version, HQ, Cognitive Failures Questionnaire, Satisfaction with Life Scales (questionnaires)	Significant reduction in tinnitus and comorbidities (insomnia, depression, hyperacusis, cognitive failures) and a significant rise in life quality	I	Yes
Ruan Q, 2021 [34]	-Age ≥ 58 y -With/without frailty -With/without HL -With/without tinnitus	No history of disability, cophosis, and vision loss	429 patients ≥58 y	Longitudinal cohort study	To study whether cognitive frailty is associated with HL and tinnitus	To assess MCI: with executive and attention domain (TMT A and B); language domain (BNT and animal list generation); memory domain (HVLT-R) (questionnaires)	Cognitive frailty patients had higher risks of severe HL and tinnitus. Cognitive impairment in tinnitus patients involved executive, memory, and attention domains; altered processing speed	II	Yes/No

y = years, HL = hearing loss; MCI = mild cognitive impairment; K-PHQ-9 = Korean version of the Patient Health Questionnaire-9; K-IADL = Korean version of the Lawton instrumental activities of daily living scale; MoCA-K = Korean version of the Montreal Cognitive Assessment; THI = Tinnitus Handicap Inventory; HADS = Hospital Anxiety and Depression Scale; MMSE = Mini Mental State Examination; HL = Hearing Loss; TMT A and B = Trail Making Test; BNT = Boston Naming Test; HVLT-R = Hopkins Verbal Learning Test, Revised.

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
