# Peer review of "Tinnitus and Neuropsychological Dysfunction in the Elderly: A Systematic Review on Possible Links"

_jcm, 2021, doi:10.3390/jcm10091881_

Round 1

Reviewer 1 Report

Although the topic of the review is very interesting, there are lots of references missing in the introduction and the entire review methodology is missing. It is possible that the authors did a sound systematic review, but it is impossible to know when the review methodology is not reported.

Author Response

Dear Reviewer,

Please find enclosed the revised version of the article titled ”Tinnitus and neuropsychological dysfunction in the elderly: a systematic review on possible links” The text has been modified according to your indications. Changes have been highlighted in red in the new text.  The authors wish to thank the Reviewer 1 for his/her kind comment and suggestion. We apologize for several mistakes in the old version of paper. We believe that the manuscript has been significantly improved. Namely, a systematic review on tinnitus and neurocognitive in the elderly has been performed, data have been presented in two new paragraph in the result section and discussion has been re-written. A new version of method section has been added in order to describe the systematic review. The title has been consequently modified. We hope that this revised version of the manuscript is now suitable for publication and we hope of a favorable response.

Reviewer 2 Report

The article “Tinnitus and cognitive dysfunction: a review on possible links” by Malesci et al. attempts to review literature that associates the presence of tinnitus with cognitive dysfunction. Overall, I do not disagree with the content in the article. However, the presentation of the information is very difficult to follow and compounded by many typographic and grammatical errors. The review article hold potential to organize the content of this literature in a single location which would provide a single hub of information for investigators. To be considered for publication I recommend a major revision of the manuscript with an emphasis on improving the language, clarity and organization. Below I provide some suggestion for completing this.

  • The article jumps to often jumps around between the impact that tinnitus has on patients, the pathophysiology and potential cognitive deficits. Focus the article by just describing one aspect at a time that is associated with corresponding headers.
  • The section on cognitive dysfunction and tinnitus is extremely muddy. Cognition is a highly complex phenomenon. The authors such discuss tinnitus impact on each aspect of cognition individually. Separating the text into sub sections would be helpful (e.g., Tinnitus and memory, tinnitus and attention, tinnitus and executive function etc.)
  • The introduction again is presented in a relatively disorganized manner. Instead of restating much of the introduction throughout the rest of the text, shorten the introduction to remove redundancy and make it easier for the reader to understand the problem being framed. As it stands the introduction is about ½ of the text.
  • The section on the pathophysiology of tinnitus is not specific to the purpose of the review. Indeed, peripheral dysfunction leads to enhanced spontaneous activity along the central auditory pathway, and this is considered by many to underly the phantom perception of tinnitus. However, there is a host of studies that have investigated the effect of hearing loss and tinnitus on non-classical auditory systems that are not mentioned. These are particularly important to support the authors hypothesis that tinnitus leads to cognitive dysfunction.

Text edits through introduction below (although there are several more throughout the body of the manuscript)

Abstract

Line 13- “occurs as an isolated idiopathic symptom and is commonly associated with hearing loss caused by noise exposure or presbycusis”

Line16 – “tinnitus neuro pathophysiology and potential relationships with cognitive functions.                    

 Line 20- literature provides evidence for the association between tinnitus and executive attention and working memory.

Introduction

Line 29 – tinnitus, the perception of sound in the absence of an external stimulus

Line 33- accompanied by hearing impairment, especially from excessive noise exposure

Line 36 – need citation for manières disease cause of tinnitus

Line 38 – odd use of the phrase “on the other hand” make it a new sentence.

Line 40 – Actually the opposite is true, most research suggests that tinnitus results from aberrant adaptation to a loss of neural output from the cochlea, not a failure to adapt.

Line 45- should not be deprivation of both excitatory and inhibitory, either excess excitation or loss of inhibition is most commonly thought to result in increased spontaneous activity thought to underly tinnitus. You say it correctly following.

Line 51 – indeed tinnitus HAS

Line 85 authors TO

Line 103 – and non-classical auditory brain regions

Line 103- Although tinnitus can be perceived in patients … this sentence is misleading, in fact tinnitus likely occurs from central abnormalities in neural processing. It may be provoked by peripheral damage, but cause of tinnitus perception is central.

Author Response

Dear Reviewer,

We are very grateful to the Reviewer 2 for his/her valuable comments. Please find enclosed the revised version of the article titled ”Tinnitus and neuropsychological dysfunction in the elderly: a systematic review on possible links”. The text has been modified according to your indications. Changes have been highlighted in red in the new text.  The authors wish to thank the Reviewer 2 for his/her kind comment and suggestion. We apologize for several mistakes in the old version of paper. We believe that the manuscript has been significantly improved. We differently organized the work focusing on psychological aspects and cognitive dysfunction in older patients affected by tinnitus. We have shortened the introduction as you asked and removed many redundances. We furthermore deleted the pathophysiology of tinnitus to make the article easier to read and understandable. The title has been consequently modified. We hope that this revised version of the manuscript is now suitable for publication and we hope of a favorable response.

Reviewer 3 Report

Authors discussed possible links between tinnitus and cognitive dysfunction by citing ample references. Are there any ongoing research that tries to address a clinical question whether treatment for tinnitus would alleviate cognitive dysfunction?

Author Response

Dear Reviewer,

We are very grateful to the Reviewer 3 for his/her valuable comments. Please find enclosed the revised version of the article titled ”Tinnitus and neuropsychological dysfunction in the elderly: a systematic review on possible links”. The text has been modified according to your indications. We have found diverse articles in which tinnitus treatment would alleviate cognitive dysfunction, and added one of them which has targeted the focus on the elderly. A comment on this topic and a new reference has been added. We hope that this revised version of the manuscript is now suitable for publication and we hope of a favorable response.

Round 2

Reviewer 1 Report

Dear authors,

Thank you for your revision of this work. Although it has improved a lot since the previous version, there are currently still many typos in the revised text. Additionally, the results section lacks focus and includes too much discussion and the results of the risk of bias and level of evidence assessment are not reported.

Author Response

We are very grateful to the Reviewer for it/her valuable comments and suggestions. We believe that quality of paper has been significantly improved thanks it/her helpful revision. Manuscript has been reviewed to address the Reviewer concerns and all  comments were taken into consideration in the new version of manuscript. Namely, the results section have been focused on the literature reports accordingly to the systematic revision and limitations of revision have been added as well as tables have been modified and details on level of evidence assessment are now clearly reported.

Reviewer 2 Report

The manuscript is now mostly acceptable. However, I still find many instances throughout the text to be cumbersome to read due to awkward wording, grammar and typographic errors. Perhaps consulting an native english speaker to review the text would be helpful. 

Author Response

We are very grateful to the Reviewer for it/her valuable comments and suggestions. We believe that quality of paper has been significantly improved thanks it/her helpful revision. Manuscript has been reviewed to address the Reviewer concerns. Namely, a deep revision of English grammar has been done and we apologize for many typographic errors because of the use of different software. We believe that all typos have been corrected but some of these could be remained and we are able to correct the new version or the proof in case the paper will be accepted for publication.